# The Effect of Active Plus, a Computer-Tailored Physical Activity Intervention, on the Physical Activity of Older Adults with Chronic Illness(es)—A Cluster Randomized Controlled Trial

**DOI:** 10.3390/ijerph17072590

**Published:** 2020-04-10

**Authors:** Esmee Volders, Catherine A. W. Bolman, Renate H. M. de Groot, Peter Verboon, Lilian Lechner

**Affiliations:** 1Faculty of Psychology, Open University of The Netherlands, 6419 AT Heerlen, The Netherlands; catherine.bolman@ou.nl (C.A.W.B.); peter.verboon@ou.nl (P.V.); lilian.lechner@ou.nl (L.L.); 2Faculty of Educational Sciences, Open University of The Netherlands, 6419 AT Heerlen, The Netherlands; renate.degroot@ou.nl; 3Nutrition and Translational Research in Metabolism (School NUTRIM), Maastricht University, 6200 MD Maastricht, The Netherlands

**Keywords:** ageing, accelerometer, self-report, randomized intervention, chronic disease, physical activity promotion, eHealth

## Abstract

eHealth interventions aimed at improving physical activity (PA) can reach large populations with few resources and demands on the population as opposed to centre-based interventions. Active Plus is a proven effective computer-tailored PA intervention for the older adult population focusing on PA in daily life. This manuscript describes the effects of the Active Plus intervention (*N* = 260) on PA of older adults with chronic illnesses (OACI), compared to a waiting list control group (*N* = 325). It was part of a larger randomized controlled trial (RCT) on the effects of the Active Plus intervention on cognitive functioning. OACI (≥65 years) with at least one chronic illness were allocated to one of the conditions. Intervention group participants received PA advice. Baseline and follow-up measurements were assessed after 6 and 12 months. Intervention effects on objectively measured light PA (LPA) and moderate-to-vigorous PA (MVPA) min/week were analysed with multilevel linear mixed-effects models adjusted for the clustered design. Intervention effects on self-reported MVPA min/week on common types of PA were analysed with two-part generalized linear mixed-effects models adjusted for the clustered design. The dropout rate was 19.1% after 6 months and 25.1% after 12 months. Analyses showed no effects on objectively measured PA. Active Plus increased the likelihood to perform self-reported cycling and gardening at six months and participants who cycled increased their MVPA min/week of cycling. Twelve months after baseline the intervention increased the likelihood to perform self-reported walking and participants who cycled at 12 months increased their MVPA min/week of cycling. Subgroup analyses showed that more vulnerable participants (higher degree of impairment, age or body mass index) benefitted more from the intervention on especially the lower intensity PA outcomes. In conclusion, Active Plus only increased PA behaviour to a limited extent in OACI 6 and 12 months after baseline measurements. The Active Plus intervention may yet be not effective enough by itself in OACI. A blended approach, where this eHealth intervention and face-to-face contact are combined, is advised to improve the effects of Active Plus on PA in this target group.

## 1. Introduction

Older adults with chronic illnesses form a large part of society [1]. In general, their physical activity (PA) levels are low [2]. This is associated with increased health problems and decreased cognitive functioning [3]. Therefore, it is important to increase PA levels of older adults with chronic illnesses. PA programs for this population exist, but attendance and results vary [4,5]. In the present manuscript, the effects of an already proven-effective computer-tailored PA stimulating intervention on PA behaviour in older adults with chronic illnesses (OACI) is examined. This is a secondary analysis of a larger study on the effects of this proven-effective PA intervention on cognitive functioning. 

Almost 85% of the older adults in developed countries have at least one chronic illness, and around 60% suffer from multiple chronic illnesses [1,6,7]. However, these chronic illnesses themselves might be not the only issue; the functional limitations and mobility restrictions older adults experience as a result of these chronic illnesses provide additional problems [8]. Examples of these functional limitations and mobility restrictions are: (1) having difficulties with walking; (2) getting into a bed or a chair; or (3) stair climbing. These problems are highly prevalent and have a negative influence on the ability to maintain their activity levels [9,10]. As a result, these functional limitations and mobility restrictions impede the ability to have an independent life and can lead to poorer health-related quality of life (HRQoL) [9,11,12].

Increasing PA is a proven-effective strategy for both prevention as well as treatment of chronic illness [13,14,15,16]. PA has other health benefits as well, such as weight control, strengthening of muscles and bones, and improvements of physical and cognitive functioning, mental health and HRQoL—all factors negatively affected by chronic illnesses [16,17]. Despite these beneficial effects of PA, older adults are the least physically active age group, especially when they suffer from chronic illnesses [2,11]. Most of them do not reach the recommendation of 150 min per week of moderate-to-vigorous physical activity (MVPA) supplemented with muscle, bone, and balance improving activities at least twice a week [18,19]. This may be due to the many PA-related barriers (i.e., fatigue, pain) experienced by older adults with chronic illnesses [2,17]. However, meta-analyses by Bullard et al. [17] and de Vries et al. [10] suggest that adherence to the guidelines for PA (≥150 min of MVPA/week) is highly feasible and effective among chronic disease populations. 

Besides MVPA, the health benefits of light-intensity physical activity (LPA) have been established as well. Objectively measured light physical activity (LPA) is associated with lower all-cause mortality risk and improved cardiometabolic risk factors and cognitive functioning [20,21,22]. LPA is perhaps more feasible for OACI than MVPA [23]. Consequently, it is suggested that, next to MVPA, LPA should be taken into account when evaluating PA [24]. 

Although a few PA programs exist for OACI (e.g., Coach2Move [25], Strong-for-Life [26], Life-P [27]), most programs are not easily accessible, and often only reach already active older adults [5]. They usually take place at a research site, gym, or physical therapist, which OACI need to visit 1–3 times per week. These programs are generally face-to-face and offer detailed and intensive supervision [17]. Yet, these programs are also highly demanding, making it more difficult for OACI to adhere to these programs, especially on long term [28]. However, not only clinic-based programs exist for OACI, but there are also proven effective home-based programs as described by Duijts et al. [29], Gary et al. [30] and Lee et al. [31]. These programs generally provide participants with a personalised PA program to be executed at home. First, participants are taught by a research nurse how to correctly perform the aerobic exercise program and how to adjust the exercise prescription. Home-based programs typically provide more autonomy (e.g., more choices regarding training schedule and fewer transportation-related barriers) [17]. Nevertheless, these existing programs for OACI are commonly focussed on physical functioning instead of physical activity. In other words, these programmes are more focussed on improving the capacity to execute habitual daily activities such as stair climbing than on having an active lifestyle with higher amounts of low, moderate and vigorous PA. Besides, a meta-analysis showed that offering exercise or PA in a program is not enough to stimulate OACI to become more physically active on their own in their daily lives outside of the gym [10]. 

Based on this knowledge, the computer-tailored PA stimulating intervention Active Plus was developed for people aged over 50 years [32,33]. At a later stage, the eHealth program was adapted to a more elderly (≥65 years) population who often suffered from chronic illnesses [34]. Active Plus participants receive 3 personalised PA advice letters (online or print delivered) in 4 months. Previous research in people aged over 50 years showed that the Active Plus group was 1.5 h per week more active at moderate to vigorous intensity after 1 year compared to controls [35], even in older adults with impaired mobility [36]. Although a recent study showed positive effects of the adapted Active Plus intervention on PA in single older adults (≥65 years) with physical impairments 3 months after baseline, no effects were found after 6 months [37]. However, this concerned an implementation study without a control group, making it impossible to draw definite conclusions. 

In this paper, the effects of the Active Plus intervention on PA in OACI are examined. Although previous studies already showed the effectiveness of the Active Plus intervention on PA in the general older adults (≥50 years) population [35], solely self-reported PA measures were used to assess intervention effects. However, self-reported PA questionnaires are known for their over-reporting of PA due to social desirability and recall accuracy [38]. Furthermore, not all intensities of PA are validly assessed by questionnaires, in particular, light to moderate PA, which is the intensity of PA older adults are most likely to engage in [39]. Therefore, in the present study, we also included objectively measured PA. The use of accelerometers in the assessment of PA became more user friendly [40], and past research suggests that it captures the quantity and intensity of PA behaviour more accurately [41]. 

Nevertheless, accelerometers too have disadvantages. Insight in which specific PA activities a person performs cannot be derived from accelerometer data [42]. For example, solely on accelerometer data from a single device, it is still impossible to discriminate between sitting and standing. Furthermore, depending on the attachment site (hip versus upper leg or wrist), some accelerometers are not able to detect all kind of movements such as upper/lower body movement or stationary movement, while these behaviours are common in older adults during gardening, household chores, and cycling. Cycling in particular is a frequent PA activity in The Netherlands [43]. In addition, most accelerometers are also not able to assess water-based activities, as not all accelerometers are waterproof [41]. Therefore, it is recommended to both use objectively measured PA with an accelerometer and subjectively measured PA with a self-report PA questionnaire in assessing PA intervention effects [44].

The present study will be one of the first studies to assess computer-tailored PA intervention effects on PA behaviour in a broad sense. We evaluate the effects of the Active Plus intervention in OACI on objectively measured LPA and MVPA, and on the likelihood to perform common PA activities and MVPA minutes per week during these activities assessed with a self-report questionnaire. As the intervention was aimed at increasing PA, it is hypothesized that the intervention group increased both their objectively measured and self-reported PA more than the waiting list control group. Although the intervention is individually tailored, it might be that not all subgroups of participants respond similarly to the intervention. Therefore, we examine in an exploratory way whether the effects differ for the degree of impairment, age, gender, body mass index, educational level, and marital status. 

## 2. Materials and Methods 

### 2.1. Study Design, Setting and Population

The Active Plus study was a clustered two-group randomized controlled trial (RCT) with a waiting list control group with assessments at baseline, 6 and 12 months. This RCT primarily focused on the effect of the Active Plus intervention on cognitive functioning. However, the present study involved the secondary analysis on the dataset. It concerns the evaluation of the effects of the intervention on PA for OACI. Ethical approval for the study was granted from the Research Ethics Committee (cETO) of the Open University and the trial is registered in the Dutch Trial Register, protocol number NL6005. The study was conducted following the Declaration of Helsinki. An extensive rationale and description of the general study protocol is published elsewhere [45]. 

Seven Dutch municipalities agreed to participate. Each municipality selected two comparable residential areas or neighbourhoods based on their social-economic status as derived from https://www.waarstaatjegemeente.nl. These areas were randomly assigned [46] to either the experimental group or the waiting list control group, so each municipality both had an intervention group residential area and a control group residential area. Municipalities selected between 250–2000 names and addresses of independently living men and women aged 65 years or older per residential area. Participants were recruited from February to July 2018 by an invitation letter from their municipality including an information letter with the study content and an informed-consent that could be returned to the researchers. Participants did not know there were two groups (intervention and control) being investigated. Eligible participants had to be 65 years or older, be fluent in Dutch, and suffer from at least one chronic illness that affects mobility (e.g., musculoskeletal and back disorder, chronic obstructive pulmonary disease (COPD), rheumatism, osteoporosis, chronic heart disease) or other physical conditions (e.g., visual or hearing impaired) that may affect mobility. Participants with severe self-reported cognitive problems or wheelchair users were excluded from the study. Participants had to be able to walk at least 100 m, possibly with the help of a rollator or walking stick. All participants provided written informed-consent. 

#### 2.1.1. Procedure

Figure 1 displays the flow diagram of the study. The measurements of cognitive functioning (CF) are outside the scope of the current study. Participants in both conditions were asked to wear an accelerometer for 7 consecutive days on their right hip. All participants received both a paper-based (with a prepaid return envelope) and an online questionnaire (integrated into the project website: www.actief-plus.nl) with the choice to fill out their preferred format within two weeks. So the objective PA measurement preceded the subjective measurement. Thereafter, the 4-month lasting intervention commenced for the experimental group. Six and 12 months after the first accelerometer measurement, participants wore that device again and completed a questionnaire following the same procedure as the baseline measurement. After the final assessment (i.e., after 12 months) participants in the control group received access to the Active Plus intervention (i.e., waiting list control group).

#### 2.1.2. Intervention Group

The original Active Plus intervention is a proven effective computer-tailored intervention that is systematically developed using the intervention mapping protocol, and based on various theoretical models such as the theory of planned behaviour [47], precaution adoption process model [48], integrated model for change (I-Change Model) [49], and self-regulation theory [50]. The intervention intended to take up and sustain PA behaviour by influencing (pre-/post-) motivational factors, such as awareness, intention, self-efficacy and action planning. These factors have been shown to impede or facilitate PA behaviour. The original Active Plus intervention was aimed at the general population of adults aged 50 years or over [32]. 

Boekhout et al. [34] adapted the intervention using the intervention mapping protocol for a chronically ill older adult (≥65 years) population. Based on a literature study, focus groups with the target population, and interviews with experts, it was concluded that the general determinants for PA in OACI were quite identical to general older adults. However, some specific determinants had a larger influence on PA. For example pain, fear of injury, and lack of energy proved to be more important barriers. Therefore, the tailored messages were extended and enhanced to align them with the needs and requirements of OACI. In the present study, the intervention was tailored to two extra common chronic illnesses (neuromuscular and vascular disorders), and information on the risks of sedentary behaviour and benefits of PA for cognitive functioning were extended. A more detailed description of the intervention content and the process of adaption on the original and adapted Active Plus intervention can be found elsewhere [32,34,45].

Participants in the intervention group received advice on three occasions, both online on a secured website and paper (via a letter by mail) (see Figure 1), tailored to the answers they gave in the questionnaires that they filled in. The first and second personal advice were based on the baseline questionnaire and were received respectively within two weeks of filling in this questionnaire and two months after filling in the baseline questionnaire. Three months after the baseline questionnaire a follow-up questionnaire was conducted and used to compose the third advice with feedback on progress in PA behaviour and relevant determinants. Participants received their third advice within two weeks after completing the follow-up questionnaire. 

As mentioned, each advice gave tailored information on PA and presented tools on how to implement PA in daily life, especially focussed on older adults with chronic illness(es). The exact content of the advice depended on the participants’ characteristics (e.g., age, gender, and presence of chronic illness), psychosocial characteristics/motivational constructs, their current PA behaviour, and to what extent they were willing to alter their behaviour (as derived from the answers on the questionnaires). The website and advice also included additional information on local PA possibilities (e.g., walking or cycling routes in their neighbourhood or local sports clubs), as well as a user forum, and examples of PA exercises. 

#### 2.1.3. Waiting List Control Group

Participants allocated to the waiting list control condition had no access to the intervention and received their usual care. After the 12-month study period, they gained access to the Active Plus intervention and received their personalised PA advice. 

### 2.2. Outcomes

Objectively measured LPA and MVPA and self-reported MVPA behaviour during common PA activities were assessed at baseline, and after 6 and 12 months. 

#### 2.2.1. Physical Activity (PA) Outcomes

PA was objectively measured using the ActiGraph GT3X-BT (ActiGraph, Pensacola, FL, USA). Participants wore the accelerometer on an elastic belt on their right hip for 7 days. During the night participants were not obliged to wear the device. Participants were asked to remove the devices while showering or swimming. Data were downloaded and analysed using ActiLife software with the low-frequency extension on [51]. Valid measurements contained at least 4 days of data with at least 10 h of wear time per day [52]. Non-wear periods were identified with the Choi algorithm and eliminated from the analyses [53]. The Choi algorithm identifies 90 min of consecutive zero counts as non-wear time, which may be interrupted by a maximum of 2 min of non-zero counts. To distinguish between light, moderate and vigorous PA the software used data from 3 axes based on 60 s epochs and Freedson-VM cut-off points, developed by Sasaki [54], and the cut-off points developed by Aguilar-Fariaz [55]. Outcome measures in the present study are the minutes spent in LPA per week and minutes spent in MVPA per week. 

Self-reported PA was measured using the validated Short Questionnaire to Assess Health Enhancing Physical Activity (SQUASH) [56], assessing activities regarding household, leisure time and sports. Questions regarding PA at work were deleted from the questionnaire because our target population is normally retired in The Netherlands at that age. For each activity the frequency, the duration and the intensity were administered. The SQUASH questionnaire classifies PA into light (metabolic equivalent (MET) <3.0), moderate (MET 3.0–5.9), and vigorous (MET > 6) [57]. The scoring manual is used to calculate these constructs and to exclude any extreme values. The SQUASH has reasonable reliability (ρ = 0.58) and validity (ρ = 0.45) opposed to an accelerometer [56]. Outcome measures in the present study are the likelihood of performing and minutes of MVPA during household activities (e.g., cleaning and cooking), walking (during leisure time and transportation), cycling (during leisure time and transportation), gardening (during leisure time and (volunteer) work), work in/around the house (e.g., DIY and small repairs during leisure time and (volunteer) work), and sports activities (e.g., swimming, fitness). 

#### 2.2.2. Other Relevant Measures

Several demographic factors such as age, gender, education, marital status (living together with a spouse or living single), body mass index (BMI), and presence of comorbid conditions, are known to influence PA behaviour [58]. Therefore, these factors were assessed at baseline. Educational level is categorized into low (i.e., primary, basic vocational, or lower general school), moderate (i.e., medium vocational school, higher general secondary education, and preparatory academic education), or high (i.e., higher vocational school or university level) according to the Dutch educational system. BMI is defined as the body mass divided by the square of body height. The degree of impairment is measured with a self-report questionnaire [34]. The participant stated for 14 common chronic illnesses (i.e., cardiovascular, osteoarthritis) and physical conditions (i.e., hearing or visually impaired) to what degree he/she is limited in his/her PA behaviour by one of the illnesses mentioned or by another illness not mentioned. For each chronic illness, the participant scores the degree of impairment on a 5-point scale ranging from 0 = not applicable, 1 = not at all/hardly, 2 = a little, 3 = very, to 4 = extremely. Consequently, degree of impairment is computed into 3 categories following the next rules: (1) Little impaired: a maximum score of 1 on at least one question, (2) Medium impaired: a maximum score of 2 on at least one question, (3) Very impaired: at least a score of 3 or 4 on at least one question. 

#### 2.2.3. Sample Size and Statistical Power

Sample size calculations are based on the primary outcome measures of the overall Active Plus project, namely cognitive functioning. Based on the effect sizes of earlier intervention studies [59], we estimated a difference in effect size (ES) of 0.3 for CF between the intervention and control group. The needed sample size has to be inflated to take account of the multilevel design. Therefore, an estimate of intra-cluster correlation (ICC) is used, based on the ICC of the previous Active Plus projects (ICC < 0.01). Statistical power analysis using G*Power (Franz Faul, Universität Kiel, Kiel, Germany) [60] (ES = 0.30; power = 0.80; ICC = 0.01) showed that 190 participants per group were required. Based on our previous study [35] we expected a 30% dropout rate at 12 months. We, therefore, needed 270 participants to be enrolled per group at baseline. For the effects of Active Plus on PA, we expected a comparable ES of at least 0.3 [35].

#### 2.2.4. Statistical Analyses

Baseline differences between conditions were tested with a χ2 test for categorical variables, a Mann–Whitney U-test for continuous variables with skewed distributions, and a one-way analysis of variance (ANOVA) for normally distributed continuous variables. To assess predictors of dropout at 6 and 12 months, logistic regression with condition, baseline outcome measures, demographics, and degree of impairment regarding chronic illnesses was performed and odd-ratios (OR) are noted. 

Linear mixed-effects models were used to assess intervention effects on LPA and MVPA measured with an accelerometer. The MVPA outcome seemed to be skewed and analyses led to non-normally distributed residuals, therefore MVPA was 10log transformed. For the highly skewed and zero-inflated self-report PA outcomes a two-part generalized linear mixed-effects model was applied to predict: a) participation in a specific PA activity (0 = no, 1 = yes), b) the duration of MVPA during these specific PA activities. The two-part model used a binomial distribution with the logit link function to model the zero versus non-zeros for participating in a specific PA activity over time and the gamma distribution for the skewed continuous data indicating the change in the amount of MVPA during a specific PA activity over time. The latter approach is recommended to account for the highly skewed and zero-inflated distribution of PA-related outcomes [61,62].

With participants originating from different municipalities, it was expected their data was clustered. Therefore, we applied multilevel analyses with participants nested in municipalities, with level 1 being the different time points, level 2 the subject, and level 3 the municipality. The analyses revealed that the ICC for MVPA was 0.07. Consequently, three-level analyses were performed for all outcomes to assess intervention effects. Time, group, and the interaction between time and group were added to the models as fixed effects to assess intervention effects over time. Intervention effects between intervention group and control group were compared between baseline and 6 months follow-up and between baseline and 12 months follow-up. All models were fitted using the maximum likelihood procedure. For all analyses age, gender, educational level, marital status, BMI, and degree of impairment were added as covariates. Continuous variables were standardized. Furthermore, confidence intervals (CI) were calculated for all outcomes. Analyses were conducted on an intention-to-treat basis without any ad hoc imputation [63]. 

Exploratory differences regarding intervention efficacy were assessed for degree of impairment, age, gender, educational level, marital status, and BMI. Three-way interaction terms (time × group × covariate) of significant covariates were added to the model. When a three-way interaction term was significant, subgroup effects were examined by repeating the analyses. In these multilevel analyses, the three-level data structure was applied again. Subgroups were defined by the categories of the covariates for categorical variables. For the continuous variable BMI the groups were split by obese or non-obese (limit at 30 kg/m^2^) and for age, the limit was at 80 years or older. Since interaction terms have less power, the significance levels were set to *p* < 0.10 for the interaction term [64]. Significance levels for all other analyses were set to *p* < 0.05. All analyses were conducted using R [65]. 

## 3. Results

### 3.1. Study Population

In total 623 participants provided informed consent for the study and were included in the study. Thirty-eight of them already withdrew from the study before baseline, resulting in 585 participants with some baseline data (Figure 1). Their mean age was 74.5 (±6.4) years with almost equal numbers of male and female participants. The majority of the participants were living together with a spouse (80.7%), and 51.2% were low-educated (i.e., primary, basic vocational, or lower general school). Most participants (46.1%) were medium impaired. The most frequent chronic illnesses participants suffered from and impaired PA were osteoarthritis (51.7% of all participants), vascular diseases (44.6%) and heart diseases (37.2%). Participants suffered from an average of 3.5 chronic illnesses or physical impairments. As shown in Table 1, no significant baseline differences were found between the intervention and control group. The drop-out rate was 19,1% (112/585) at 6 months and 25,1% (147/585) at 12 months. Drop-out at both 6 and 12 months after baseline was more likely for participants in the intervention group (6 m: OR = 6.85, 95% CI = 3.78;13.07, *p* ≤ 0.001; 12 m: OR = 2.95, 95% CI = 1.87;4.72, *p* ≤ 0.001) and elderly participants (6 m: OR = 1.09, 95% CI = 1.04;1.15, *p* ≤ 0.001; 12 m: OR = 1.07, 95% CI = 1.03;1.11, *p* = 0.001). Most control group participants (30 out of 64 dropouts) dropped out of the study because of being too ill to continue, while intervention group participants mostly (48 out of 89 dropouts) dropped out because they lost interest. Especially during the first part of the intervention, participants dropped out when they had to fill in a follow-up questionnaire needed to compose the third advice. 

### 3.2. Intervention Effects

Although raw means (Table 2) seem to suggest that the intervention group improved their scores on objectively measured MVPA, and self-reported household activities, walking, cycling, gardening and sports activities as opposed to the control group, tests for significance only showed limited effects. The results are shown in Table 3. 

Six months after baseline results show that participants in the Active Plus group were more likely to perform self-reported cycling (Coeff. = 1.12, *p* = 0.01) and gardening (Coeff. = 0.77, *p* = 0.04) after adjusting for covariates. For those individuals engaging in the specific self-reported PA behaviours the intervention was effective in increasing the amount of MVPA during cycling (Coeff. = 0.34, *p* = 0.005) after adjusting for covariates. No significant differences were found in objectively measured MVPA (Coeff. = 0.04, *p* = 0.20) or LPA (Coeff. = −18.47, *p* = 0.68). 

Twelve months after baseline results indicate a significantly higher likelihood of participants in the Active Plus group perform self-reported walking after adjusting for covariates (Coeff. = 0.40, *p* = 0.04). For those individuals engaging in the specific self-reported PA behaviours, the intervention was effective in increasing the amount of MVPA during cycling (Coeff. = 0.28, *p* = 0.02) after adjusting for covariates. No significant differences were found in objectively assessed MVPA (Coeff. = −0.00, *p* = 0.93) or LPA (Coeff. = 2.25, *p* = 0.96). 

### 3.3. Moderation of Effects

Although the intervention is individually tailored, it might be that not all subgroups of participants respond similarly to the intervention. Therefore, to further explore the efficacy of the intervention, analyses for subgroups were performed. These exploratory analyses (Table 4) showed that the significant effect of the intervention (intervention group vs. control group) on the likelihood to perform self-reported walking behaviour was moderated by the degree of impairment participants experienced. Participants whom were only little impaired did not improve significantly at 6 (Coeff. = −2.26, *p* = 0.18) or 12 months after baseline (Coeff. = 12.07, *p* = 0.98), while participants whom were medium impaired or very impaired, were significantly more likely to perform walking behaviour at both 6 months (Coeff. = 0.87, *p* = 0.04) and borderline significant at 12 months (Coeff. = 0.81, *p* = 0.054).

Despite there being no intervention effects on accelerometer assessed LPA in the complete population, the intervention was borderline effective in increasing LPA in OACI with a BMI of 30 kg/m^2^ and higher (Coeff. = 169.18, *p* = 0.096) 12 months after baseline, but not in OACI with a BMI lower than 30 kg/m^2.^ (Coeff. = −40.91, *p* = 0.43) as opposed to the control group. 

In addition, the intervention effect on the likelihood of doing self-reported odd-jobs was moderated by age. More elderly participants in the intervention group (≥80 years) were significantly more likely to perform odd-jobs (Coeff. = 2.98, *p* = 0.02) 12 months after baseline as opposed to the control group, but participants under 80 years were not (Coeff. = −0.16, *p* = 0.70). 

Another moderator was educational level. Participants in the intervention group with a medium educational level who engaged in household activities did this for longer per week (Coeff. = 0.44, *p* = 0.01) at 6 months after baseline, than did participants with a low educational level (Coeff. = 0.11, *p* = 0.32) or a high educational level (Coeff. = −0.14, *p* = 0.31) as opposed to the control group.

## 4. Discussion

The current study assessed the effects of the computer-tailored Active Plus intervention on objectively measured and self-reported PA in older adults with chronic illness(es). Additionally, it explored the effectiveness of the intervention in subgroups. 

Overall, the effects of the Active Plus intervention on PA in OACI were limited. The hypothesis that the intervention group would increase their objectively measured PA was not confirmed. Although at 6 months follow-up there seemed to be a positive intervention effect tendency for MVPA, there was no statistical evidence. The hypothesis that Active Plus would increase self-reported PA behaviour during specific activities was only partly confirmed. At 6 months after baseline, the intervention was effective in increasing the likelihood to perform self-reported cycling and gardening. For those individuals engaging in cycling at 6 months, the intervention was effective in increasing the amount of MVPA during cycling as well. At 12 months after baseline, the likelihood to perform self-reported walking improved significantly more in the intervention group. Besides, the intervention appeared effective in increasing the amount of MVPA during cycling at 12 months after baseline. In sum, we did not see any effects on objectively measured PA and only limited effects on self-reported PA.

However, these results are not in line with previous research on the Active Plus intervention. Peels et al. [35] demonstrated that Active Plus was effective in increasing PA in adults of 50 years or older both in short- and long term. Participants in this previous RCT had a mean age of ±63 years and only ±40% of them had a chronic limitation. Hence, both populations differ substantially and this possibly explains the limited results we found in the present study. It is possible that the Active Plus intervention is too voluntary for OACI. In addition, a more recent study on the adapted version of the Active Plus intervention had a more comparable population consisting of single older adults over 65 years with a chronic impairment in PA. This study showed limited effects of Active Plus on PA too [37]. However, this concerned an implementation study without a control group, making it impossible to draw definite conclusions. Both studies only measured self-reported PA, so our findings concerning the objectively measured PA could not be compared.

As there were differences in the way of measuring the degree of impairment in all three Active Plus studies, it is hard to conclude that an eHealth intervention like Active Plus is less effective in increasing PA in a more ill population. To our knowledge, there are no meta-analyses that studied the effect of PA promoting eHealth interventions in OACI. However, a meta-analysis by Chase [19] showed that PA interventions, including home-based interventions, tested among healthier participants had larger effects than those tested among chronically ill populations, although effectiveness varies between different groups of chronic illnesses [66]. In addition, starting levels of PA in the present study were already relatively high with a raw mean of objectively measured MVPA of 193–210 min per week. Hence, there may be less room for improvement in this population (ceiling effect). Therefore, very large effects should not be expected when studying already physically impaired older adults [10]. 

Furthermore, next to the different target populations/ samples in the three studies and the already active population in the present study, the intervention itself and design of the study could explain the limited effects found in the present manuscript. The Active Plus intervention was specifically adapted to an older and a chronically ill population with the intervention mapping protocol based on a literature study, focus groups with the target population and expert interviews. However, constructs and affected determinants of PA of the original proven-effective intervention did not change. The degree of importance of certain determinants and consequently the tailored messages did change to some extent [34]. Presumably, the tailored messages need to be fine-tuned even more to the target population or are not convincing enough to uptake PA behaviour in this specific target group.

In addition, in the present study, we saw a larger dropout of relatively older participants due to a loss of interest during the intervention period and especially at the time the follow-up questionnaire (needed to compose the third advice) was sent out. One might consider that the intervention up to that point was not what the older participants expected. Conceivably, the relatively older participants expected the intervention to involve contact moments. Furthermore, it is possible participants thought the questionnaire was too extensive. An eHealth intervention study of Van der Mispel et al. [67] concluded that in eHealth interventions extensive questionnaires should be avoided as dropout rates were higher in interventions with rather lengthy questionnaires than in interventions with an interactive character. Although an extensive questionnaire allows for more adequate tailoring, it may be possible to shorten it. However, at this point, when the follow-up questionnaire was sent out, participants already received two out of three times advice which contained most of the information available in the intervention. It is possible that participants were already satisfied at this point, and got out of the intervention what they needed and expected. This may especially be the case for online interventions [68]. A study into the appreciation of the intervention might provide more answers for why the dropout rate was higher for relatively older participants during the intervention period.

In contrast to the above, in the current manuscript, exploratory subgroup analyses suggest that more vulnerable OACI participants benefitted more from the Active Plus intervention on several PA outcome measures, especially on the lower intensity PA outcomes. Firstly, intervention participants who were more severely impaired increased the likelihood of performing walking behaviour in contrast to control group participants. In addition, participants with a BMI of ≥30 kg/m^2^ had borderline significant higher increases in LPA. Additional analyses suggest that participants with a BMI of ≥32 kg/m^2^ did increase LPA significantly. Besides, the likelihood of doing odd-jobs increased for participants of 80 years or over. Furthermore, participants with a medium level of education had improved MVPA minutes of household activities. Thus, it appears that different subgroups respond in diverse ways to specific parts of the intervention and more vulnerable participants improve mostly on the lighter intensity activities. Possibly these are better achievable in the more vulnerable population. However, none of de covariates was a moderator on more than one PA outcome measure. Therefore, these results should be taken into account with precaution. Nonetheless, the intervention could be tailored more to the specific needs of the non-responsive subgroups. 

While only the computer-tailored PA stimulating Active Plus intervention is not sufficient to increase PA in the general OACI population, a possible solution to increase the effect could be a blended approach in which this eHealth intervention and face-to-face contact are combined [69]. A blended approach could be a cost-effective solution, as it implies less costly face-to-face contact and improved feeling of self-regulation. For example, the Active Plus intervention, which contains solely personalised advice, could be combined with face-to-face contact with a physiotherapist or weekly meetings with a PA group for older adults. A blended approach is increasingly being applied in both healthcare and mental healthcare [69]. There are already some examples of blended approach interventions aimed at promoting PA in older adults [70,71]. However, only a few studies exist and results are varying. Accordingly, additional research (i.e., meta-analysis) is necessary to identify what type of intervention (i.e., web-based, supervised, blended, etc.) works best for whom and which healthcare professionals are most suitable to refer and guide OACI to the intervention and eventually guide them. 

Although we did not find any significant improvements on objectively measured PA, some improvements on subjectively measured PA were found. Most benefits were seen in self-reported cycling behaviour, as participants in the Active Plus group were more likely to perform cycling at 6 months after baseline. In addition, participants who cycled performed this behaviour for a longer time at both 6 and 12 months after baseline. While using a validated questionnaire [56], self-reported questionnaires assessing PA behaviour are known for their over-reporting [38]. Nonetheless, waist-worn accelerometers do not detect all movements such as gardening, upper body strength exercises and cycling [42]. As we found most improvements on self-reported cycling at 6 months after baseline, and this activity is difficult to measure with an accelerometer, this problem might explain not finding effects with the objective PA measurement. Certainly, because it was fall/winter at the time of the second measurement people were more likely to stay indoors at that time because of the increased rainfall and the decreased daylight [72]. The Active Plus intervention pays attention to the weather and seasonal effects, as the intervention aims to increase the self-efficacy of participants to keep a higher PA level during bad weather/colder seasons and provides options to exercise at home. It is possible that this advice has helped to increase cycling behaviour despite the colder weather, but due to the limitations of the accelerometer (which cannot measure cycling properly), we only found limited effects. Therefore, our results can be considered of value and clinical relevance. 

Some strengths and limitations should be noted. Firstly, the current study has a strong research design (RCT) in which both objective (accelerometer) and self-reported PA (questionnaire) information were assessed [44] to give more insight in the complexity of PA behaviour. Although, both measures have their strengths and weaknesses. Self-report questionnaires are known for over-reporting, whereas accelerometers do not measure certain activities properly (e.g., cycling, swimming) [42]. By assessing objectively measured LPA and MVPA and self-reported MVPA behaviour during common PA activities we tried to gain the best insight into PA behaviour. Secondly, our research population was fairly varied and therefore the generalizability with a general OACI population seems reasonable. For instance, our research sample consisted of almost equal groups of male and female participants, and a considerable part of the participants was low educated (e.g., 51%). The mean number of comorbidities (3.5) is also in accordance with numbers in the general older adult population [73] of The Netherlands, as well as BMI levels [74]. Thirdly, the statistical method we used (two-part generalized linear mixed-effects model) to analyse the self-reported PA outcomes, is not used often. Most studies apply a linear mixed-effects models, but with self-reported PA data being highly skewed and zero-inflated this is not the optimal method [61]. Finally, by conducting multilevel analyses in this study, the most accurate way of handling missing data was applied [64].

Limitations were selective dropout, statistical power for multilevel analyses, no correction for multiple testing, and no information on adherence to the intervention. Although the selective dropout (i.e., older participants and during the intervention period) may have affected our findings, this is expected to be less detrimental because of the relatively low dropout. A dropout rate of 25.1% per cent is considered low in (partly) digital health interventions [75]. Next, as the power calculation was based only on subject level analyses and not municipality level, primary and moderator analyses may have been underpowered, as the ICC was 0.07 instead of the expected <0.01. Large inclusion numbers and a relatively low dropout rate may have limited this potential problem. Besides, we performed multiple tests to show the results of this study in a broader perspective and to give a more nuanced picture of PA behaviour. However, the more tests that are done, the more likely erroneous conclusions are drawn, because the probability of a Type 1 error is increased [76]. A Bonferroni correction, however, assumes that all of the hypothesis tests are statistically independent, which is not the case in the current study and is, therefore, overly conservative. The probability of making a Type 1 error would be less than Bonferroni assumes, and the Bonferroni correction would be an overcorrection. Therefore, we did not apply a Bonferroni correction, but results should be taken into account with precaution. Finally, adherence to the intervention was not administered during this study, therefore it is not known to what extent participants read or used the Active Plus intervention advice. However, previous research on Active Plus showed that printed materials were read by more than 93% of the participants [77]. As participants were provided with both printed and online materials, reading level/intervention exposure in the current study is expected to be similar. 

## 5. Conclusions

Our findings indicated that Active Plus was only able to increase PA behaviour to a limited extent in OACI 6 and 12 months after baseline measurements. Although subgroup analyses showed that more vulnerable participants (e.g., those with higher age, weight or more impairment) seemed to benefit more from the intervention on some specific lower intensity PA outcomes, it is possible that the Active Plus intervention is not effective enough on its own. A blended approach, in which this eHealth intervention is combined with a face-to-face intervention, is recommended to improve the effects of Active Plus on PA in OACI. This, however, needs further study.

## Figures and Tables

**Figure 1 ijerph-17-02590-f001:**
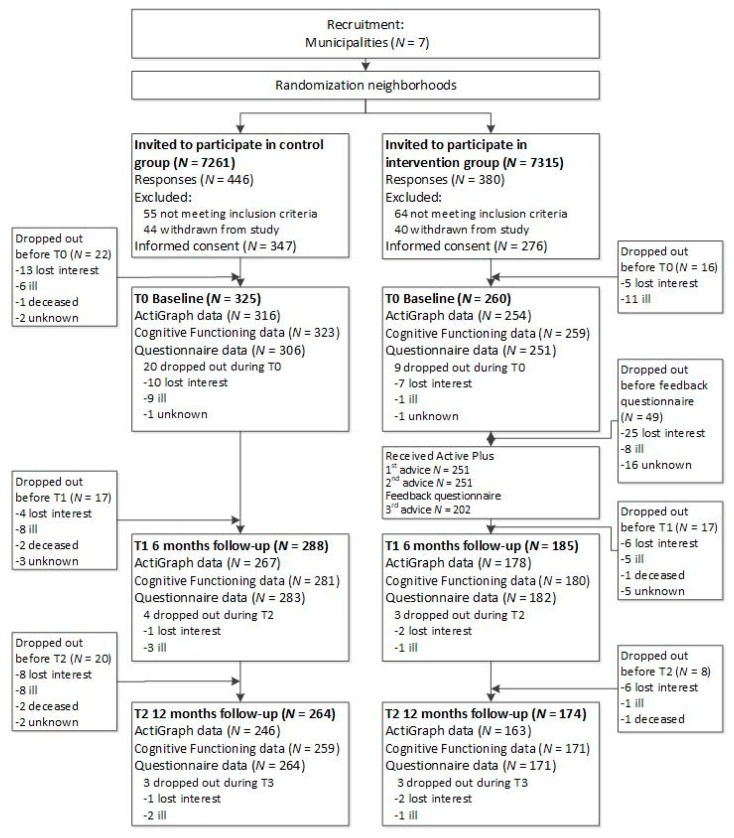
Flow diagram of the study.

**Table 1 ijerph-17-02590-t001:** Baseline participant characteristics of the control group and the intervention group.

	Control Group	Intervention Group	*p*-Value
	(*N* = 325)	(*N* = 260)	
Demographic characteristics			
Age in years, mean (SD)	74.46 (6.22)	74.20 (6.60)	0.62
Gender, *N* (%)			0.59
Male	164 (50.5%)	138 (53.1%)	
Female	161 (49.5%)	122 (46.9%)	
Marital status, *N* (%)			0.09
Living single	50 (16.6%)	56 (22.6%)	
Living together	252 (83.4%)	192 (77.4%)	
Education, *N* (%)			0.54
Low	151 (50.3%)	127 (52.3%)	
Middle	60 (20.0%)	54 (22.2%)	
High	89 (29.7%)	62 (25.5%)	
Health-related characteristics			
BMI, median (IQR) ꝉ	26.9 (24.1–29.4)	26.9 (24.4–29.8)	0.35
Degree of impairment, *N* (%)			0.39
Little impaired	34 (11.1%)	29 (11.6%)	
Medium impaired	134 (43.8%)	123 (49.0%)	
Very impaired	138 (45.1%)	99 (39.4%)	
Objective PA characteristics			
LPA, mean min/wk (SD)	2486 (641)	2494 (674)	0.88
MVPA, median min/wk (IQR) ꝉ	145 (57–290)	142 (61–261)	0.70
Self-reported PA characteristics			
Household activities			
Number of OACI performed, *N* (%)	265 (87.7%)	223 (90.7%)	0.35
MVPA, median min/wk (IQR) ꝉ	690 (360–1050)	660 (330–1170)	0.88
Walking			
Number of OACI performed, *N* (%)	209 (69.9%)	175 (70.3%)	0.99
MVPA, median min/wk (IQR) ꝉ	180 (90–360)	210 (90–360)	0.99
Cycling			
Number of OACI performed, *N* (%)	150 (50.2%)	114 (45.8%)	0.35
MVPA, median min/wk (IQR) ꝉ	180 (76.3–420)	150 (90–360)	0.64
Gardening			
Number of OACI performed, *N* (%)	155 (51.8%)	122 (49.0%)	0.56
MVPA, median min/wk (IQR) ꝉ	180 (100–360)	180 (92.5–360)	0.61
Odd-jobs			
Number of OACI performed, *N* (%)	100 (33.4%)	87 (34.9%)	0.78
MVPA, median min/wk (IQR) ꝉ	150 (60–480)	180 (67.5–405)	0.99
Sports activities			
Number of OACI performed, *N* (%)	152 (50.8%)	134 (53.8%)	0.54
MVPA, median min/wk (IQR) ꝉ	150 (90–300)	140 (75–262.5)	0.67

Abbreviations: SD, standard deviation; IQR, Inter Quartile Distance; BMI, body mass index; PA, physical activity; LPA, minutes of light physical activity per week; MVPA, minutes of moderate-to-vigorous physical activity per week; OACI, older adults with chronic illnesses. ꝉ non-normally distributed variables tested with the Mann–Whitney U test.

**Table 2 ijerph-17-02590-t002:** Raw outcomes at baseline and follow-up measurements by treatment group *.

	Measurement	Control Group		Intervention Group	
		*N*	Mean (SD)	*N*	Mean (SD)
Objective PA					
LPA min/wk	Baseline	315	2486 (641)	254	2494 (674)
	6 months	267	2374 (610)	178	2389 (595)
	12 months	246	2414 (641)	164	2455 (585)
MVPA min/wk	Baseline	315	210 (206)	254	193 (181)
	6 months	267	191 (176)	178	204 (184)
	12 months	246	206 (197)	164	200 (206)
Self-reported PA					
Household activities min/wk	Baseline	306	763 (808)	251	753 (715)
	6 months	270	672 (663)	177	747 (637)
	12 months	263	648 (701)	171	719 (638)
Walking min/wk	Baseline	306	212 (346)	251	187 (252)
	6 months	270	193 (315)	177	248 (340)
	12 months	263	185 (279)	171	216 (257)
Cycling min/wk	Baseline	306	163 (349)	251	137 (308)
	6 months	270	87 (155)	177	142 (266)
	12 months	263	135 (241)	171	155 (270)
Gardening min/wk	Baseline	306	141 (241)	251	152 (286)
	6 months	270	80 (251)	177	110 (253)
	12 months	263	149 (323)	171	186 (317)
Odd-jobs min/wk	Baseline	306	130 (350)	251	112 (266)
	6 months	270	124 (308)	177	120 (291)
	12 months	263	100 (312)	171	113 (266)
Sports activities min/wk	Baseline	306	115 (197)	251	119 (209)
	6 months	270	113 (171)	177	149 (237)
	12 months	263	109 (171)	171	131 (211)

* Summary statistics using all the available individual data at baseline and follow-up points including data consisting of 0 when participants did not perform a specific self-reported PA activity. Abbreviations: SD, standard deviation; PA, physical activity; LPA, light physical activity per week; MVPA, moderate-to-vigorous physical activity per week.

**Table 3 ijerph-17-02590-t003:** Intervention effects (Group × Time interaction) on PA outcomes for 6 and 12 months follow-up *.

		Effect After 6 Months				Effect After 12 Months			
	*N*	Coeff.	SE	95% CI	*p*	Coeff.	SE	95% CI	*p*
Objective PA									
LPA min/wk	529	−18.47	44.95	−106.6; 69.6	0.68	2.25	46.32	−88.53; 93.02	0.96
MVPA min/wk ^1^	529	0.04	0.03	−0.02; 0.10	0.20	−0.00	0.03	−0.06; 0.06	0.93
Self-reported PA									
Household activities									
Likelihood to perform ^2^	533	0.13	0.60	−1.65; 1.81	0.83	0.12	0.58	−2.09; 1.56	0.84
MVPA min/wk ^3^	505	0.10	0.08	−0.30; 0.56	0.16	0.13	0.08	−0.18; 0.57	0.08
Walking									
Likelihood to perform ^2^	533	0.68	0.40	0.01; 1.48	0.09	0.84	0.40	0.27; 1.62	**0.04^ꝉ^**
MVPA min/wk ^3^	457	0.16	0.09	−0.23; 0.66	0.08	0.15	0.09	−0.19; 0.54	0.09
Cycling									
Likelihood to perform ^2^	533	1.12	0.44	0.28; 1.93	**0.01**	0.57	0.45	−0.25; 1.43	0.20
MVPA min/wk ^3^	312	0.34	0.12	−0.10; 1.02	**0.005**	0.28	0.12	−0.13; 0.86	**0.02**
Gardening									
Likelihood to perform ^2^	533	0.77	0.38	0.09; 1.41	**0.04**	0.36	0.37	−0.23; 1.15	0.34
MVPA min/wk ^3^	328	−0.11	0.13	−0.65; 0.45	0.40	0.05	0.11	−0.39; 0.61	0.67
Odd-jobs									
Likelihood to perform^2^	533	0.14	0.38	−0.49; 0.88	0.72	0.16	0.39	−0.57; 0.92	0.67
MVPA min/wk ^3^	267	0.08	0.15	−0.53; 0.64	0.60	0.20	0.15	−0.39; 0.85	0.19
Sports activities									
Likelihood to perform ^2^	533	0.38	0.39	−0.18; 1.04	0.32	0.34	0.39	−0.27; 1.01	0.38
MVPA min/wk ^3^	360	0.04	0.09	−0.43; 0.64	0.62	0.09	0.09	−0.44; 0.66	0.33

Abbreviations: SE, standard error; CI, confidence interval; PA, physical activity; LPA, light physical activity per week; MVPA, moderate-to-vigorous physical activity per week.* Effects are reported as intervention group vs control group as the control group served as a reference group. ^ꝉ^ Number in bold print are statistically significant values (*p* < 0.05). ^1^ MVPA was 10log transformed. ^2^ Likelihood to perform a specific self-reported PA activity was analysed with a binomial generalized linear mixed-effects model with the logit link function adjusted for the clustered design. ^3^ Minutes of MVPA per week during a specific self-reported PA activity was analysed with a generalized linear mixed-effects model with a gamma distribution adjusted for the clustered design.

**Table 4 ijerph-17-02590-t004:** Moderation of intervention effects (Group × Time interaction) on PA outcomes for 6 and 12 months follow-up in subgroups *.

	Subgroup	Effect After 6 Months				Effect After 12 Months		
		*N*	Coeff.	SE	*p*	Coeff.	SE	*p*
Objective PA								
LPA min/wk	BMI <30 kg/ m^2^	410	−43.04	50.23	0.39	−40.91	51.82	0.43
	BMI ≥30 kg/ m^2^	119	74.88	99.44	0.45	169.18	101.74	0.096
Self-reported PA								
Likelihood to perform walking ^1^	Little impaired	53	−2.26	1.67	0.18	12.07	418.04	0.98
	Medium/very impaired	480	0.87	0.42	**0.04 ^ꝉ^**	0.81	0.42	0.054

Likelihood to perform odd-jobs ^1^	<80 years	420	−0.04	0.41	0.92	−0.16	0.42	0.70
	≥80 years	113	1.35	1.09	0.21	2.98	1.26	**0.02**

MVPA min/wk of household activities ^2^	Low education	257	0.11	0.11	0.32	0.13	0.11	0.24
	Middle education	106	0.44	0.17	**0.01**	0.22	0.17	0.21
	High education	142	−0.14	0.14	0.31	0.07	0.14	0.62

Abbreviations: SE, standard error; PA, physical activity; LPA, light physical activity per week; MVPA, moderate-to-vigorous physical activity per week.* Effects are reported as intervention group vs control group as the control group served as a reference group in the different subgroups. ^ꝉ^ Number in bold print are statistically significant values (*p* < 0.05). ^1^ Likelihood to perform a specific self-reported PA activity was analysed with a binomial generalized linear mixed-effects model with the logit link function adjusted for the clustered design. ^2^ Minutes of MVPA per week during a specific self-reported PA activity was analysed with a generalized linear mixed-effects model with a gamma distribution adjusted for the clustered design.

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
