# Peer review of "The Effect of Active Plus, a Computer-Tailored Physical Activity Intervention, on the Physical Activity of Older Adults with Chronic Illness(es)—A Cluster Randomized Controlled Trial"

_ijerph, 2020, doi:10.3390/ijerph17072590_

Round 1
Reviewer 1 Report
Dear author(s), I have read your manuscript with pleasure and I am enthusiastic about your performed study. Although, In my opinion your manuscript needs revisions.
This present research reports a relevant research question, although it’s a sub analysis from the major clinical trial. The research is relevant for public health and local authorities facing the issues of older adults in PA and prevention policies in upcoming years. Besides, for the field of PA research the outcome may improve interventions for other PA studies including Ehealth studies.
First remarks with first impression:
- ActivPlus; how does the adapted ActivPlus show these different outcome on PA on older adults? How does the design fit the new population? Why is the intervention and design extensively discussed, while you have previous publications? Why is the focus in de discussion on the population and statistics?
- ActivPlus explanation and theoretical construct; the interventions focussed on motivational components to adjust the PA behavior of older adults. With the intervention mapping, you showed to use different theories and concepts and use motivational factors, such as awareness, intention, self-efficacy and action planning. I was wondering why the advises are primarily focusing on the knowledge component for this target population. And I was wondering why external factors like weather/seasonal factors do get little attention in discussion while previous PA studies show other relevant factors that might attribute to the results.
As the major focus is on PA and PA behavior and the interventions were targeting on the motivation components of this behavior, I was expecting more of this ‘intervention and related constructs’ in the introduction and discussion.
- The text is very dense at some points, and therefore less readable. Which is undermining the focus and message.
- Choice to have objective and subjective PA outcome is strong.
- Additional analysis on personal aspects is a nice add, although this part does not get the right attention in results. While you do discuss it in the discussion section. My suggestion would be to delete figure 1 (or move to online as it is previously published) and add a table for the additional results.
- The abstract needs improvement in layout and focus.
So you have decided to salami slice the outcomes of the trail. Which is not actually an issue.
General: I do think the introduction/methods can be more concise, as you can refer to your previous research. At some parts too much elaboration on topics, which make the manuscript less attractive to read. As you have chosen to focus on PA, this should get the major attention. Including the factors/intervention and design aspects that may have influenced the trial results.
In my opinion the sub analysis is only explicitly clear from the design section and this needs to be clearer from title, abstract and introduction.
Strengths:
- RCT design with older adults with chronic disorders in large population based setting. Comparing an intervention with a control.
- Design of interventions; the computer-tailored design is an innovative strategy in PA interventions. Large study populations in this field are still lacking.
- Inclusion criteria; related to actual public health environment.
- Assessment techniques: objective and subjective outcome.
Weaknesses:
The quality of the presentation, clearness and elaboration on the main focus “PA” are weaknesses.
- Quality of the presentation.
With a strong design and previous published intervention, the focus of the manuscript may be improved. In my opinion the total appearance of the paper is reducing the quality of the paper. Especially as the statistical section and results are good presented.
Consistency; if you describe the measurements in methods, present them sequentially in results and tables (e.g. Table 1, vs others) - Clearness.
I would suggest to engage in more active writing on paragraph level and between sentences level, and rewrite some sections on English language. Besides, some sentences are very similar, are superfluous or seem to have similar messages. This makes the reading at some parts hard. Some examples below… - Methodological quality of the study.
To start with, the methods section is quite extensively. I would suggest to keep the necessary information, because that’s the beauty of a design study publication.
In general the extensiveness draws attention from the focus. The PA outcomes are good explained, including the statistics. I do think you have adjusted the analysis compared to your first idea and design, but you describe the decisions accordingly.
- One point of attention is the dropout; Following the flowchart there is a major drop during the intervention of 6 months. This does not get that much attention, because you describe the dropout at 12months.
- The other point of attention; you have described a cluster trial including the adjustments for cluster. This is not clearly mentioned on all statistical considerations.
Comment list:
1. Publication on secondary outcome; l.135 mentioned. Should be considered in title/abstract/intro.
2. L.36/l.42 similar messages.
3. L.46. because of , use, sentence does not read well.
4. L.48. Stay active or maintain their activity levels?. During the introduction some different terminology about physical functioning, physical activity, ADL activity and PA types are discussed. Reconsider the terminology in the introduction. L.51-l.63 for example focus on functioning, PA and ends with exercise therapy. This is quite a shift.
4. L.49. Dutch English; stand in the way; e.d. impede the ability of having
5. L.81-l83. consider reformulation. Do you consider PA a domain of PF? Be straight forward, for example it’s about habitual daily activity.
6. From l.86. You are describing the intervention. But in methods again… What’s the exact focus for the introduction?
7. From l.101 your introducing the main RQ and objectives. Be concise, and repeat this in Abstract and Discussion (efficacy? L.380-l82).
From l.123 again introducing the objectives; effectiveness. For the introduction; improve the funnel style into the main message.
8. From l. 158. Less extensive details possible. Also replacement of the design possible. See protocol. e.g. l.187; this advice on three occasions is important. The fact that’s not an online-only intervention but digital + paper intervention is important. Make choices for the redundant information and reduce the text. Previous studies on (geriatric) older patients and exercise in more frail populations show more contacts in 6 months for example. The exact contacts do not receive that much attention in the discussion later on.
9. You site and describe your previous studies a lot of times, in introduction, methods and discussion. A lot of what’s effective and what’s not, but the messages are occasionally clear and sometimes contradiction and confusing. L. 98-99 no effect vs l. 176-177 does indicate exact the opposite.
10. Intervention section: L.177 you have used IM for the adjusted version. For me it’s quite nonlogically you implemented advise on the knowledge domain for the population of chronically ill only. Additionally, from perspectives of experts/target and needs and barriers including external ones, I would expect some less information about the normal ActivPlus and some more on this adapted version. The constructs behind this version are possible more interesting to discuss in your discussion section. The differences between the interventions are possible ways to explain the differences of the results.
11. L.190. 4 months after baseline. Table says 3m+2weeks. This suggests the participants were waiting for a month for their advice. In my opinion the first actual feedback oriented ‘tailored’ advise comes quite late for a tailored intervention. And it comes 1 month after the questionnaire, which may have contributed to the dropout. The same dropout I mentioned above, which is not described that extensively. In my opinion an interesting weakness of the design and intervention. Especially if you use motivation as a construct, and do research with chronically ill. The advise and way of conducting needs more discussion.
12. L.199-208 example of extensiveness. Also website+paper is repeated from previous paragraph.
13. Figure 1 & 2: T0 T1 T2 T3 do not match.
14. Figure 2; dropout & lost of interest interestingly in table, but do not receive much attention in text.
15. Figure 2; N of assessed people does not match with N’s of assessed instruments. Not for cognitive assessments, but also not for PA assessments. Draws attention, not explained.
15. L.217. were assessed ….., remove “were taken into account” for active writing.
16. Section 2.2.2.; computed degree of impairment is based on the severity and not on the amount of diseases. So someone with multimorbidity is not very impaired following this method, but may show more issues with executing of activities and PA. Why have you chosen this method?
17. L.311. add scientific notation. (SD).
18. Table 3. Add in notes; effects are reported as …. for intervention vs control. Table can show all relevant info to read the table.
19. Explorative outcomes/moderation; 3 paragraphs might have their own table (optional)
20. In some areas of the discussion section it’s run-on sentences and readability is reduced. I would recommend to add conjunctions in the discussion section and take a recap on the English. In addition, if you take a look to the topic per paragraph, the focus of the discussion might be adjusted. This is a matter of the extensiveness of explanations.
21. L.389-l.393; subjective vs objective outcomes described. Difficult section to read. Based on the hypothesis in introduction I was expecting a statement on objective outcomes and a statement on the subjective outcomes. Final sentence is most important and is only small part of this paragraph.
22. L.416; elaboration on the population is extensive. In my opinion you have an active group of older adults. If you can only state the ceiling effects and do not question the design and intervention it’s limited. I doubt the ceiling effects, as this is not previously reported in exercise trials with more contacts and supervision. This is the part where I’m missing the theoretical construct on motivation and intervention.
23. L. 442, references?
24. L.456 seasonal/weather circumstances (important PA external factors) that do not receive that much attention. How does the timing of the measurement visits influence the results in general?
25. L 464, dropout is indeed ok. But the dropout in the intervention period is relatively higher. This is a weakness and I doubt to mention dropout as a strength.
26. L.478; sentence / words; adherence to intervention.
27. conclusions; vulnerable? The results are based on impairment and illnesses. Stimulation concept is not clear from discussion. The blended approach is a new concept for a lot of readers. Consider reformulation or strategic choice if you want to add to abstract.
28. Rewrite the abstract accordingly
Hopefully my comments and suggestions will improve the manuscript. I wish you the best in the process.
Reviewer 2 Report
The purpose of the current study was to assess the effectiveness of the computer-tailored Active Plus intervention at increasing objectively measured and self-reported PA in elderly people with chronic illness(es). The manuscript is generally well-written and the results are clearly presented.
In abstract, please explain what the RTC acronym is.
Keywords of the paper should not include words that are in the title of the manuscript.
Many gerontology professional societies discourage from using the word "elderly", and prefer the phrase "older adults". Changing this throughout the manuscript would improve the article.
Please replace references items 6, 7 and 8 with literature in English language.
Reviewer 3 Report
The manuscript "The effect of Active Plus, a computer-tailored physical activity intervention, on physical activity of elderly people with chronic illness(es) – a cluster randomized controlled trial" is a well-written contribute to the literature in the field of physical activity promotion for older adults with chronic illnesses.
The study was well structured as for length of the RCT as for measurement tools.
The only remark I can make regards the aim of the Active Plus study. The authors report that the improvement of cognitive functions is the main aim of the project, while the manuscript is focused on PA promotion. Anyway, I think that some details regarding the organizational components of the initiative and some correlation between the results obtained both in PA levels and cognitive aspects should be reported.
